Progress of circRNA/lncRNA-miRNA-mRNA axis in atrial fibrillation

Wen Jia-le 1 2
Ruan Zhong-bao 1 tzcardiac@163.com
Wang Fei 1
Hu Yuhua 2
1 Department of Cardiology, The Affiliated Taizhou People’s Hospital of Nanjing Medical University, Taizhou School of Clinical Medicine, Nanjing Medical University , Taizhou , China
2 Dalian Medical University , Dalian , China
Navaneethabalakrishnan Shobana
Electronic publication date: 2023 Dec 19
Publication date: 2023
Volume: 11
Electronic Location ID: e16604
Received 2023 Jun 29; Accepted 2023 Nov 15
Copyright: © 2023 Wen et al.
Copyright year: 2023
Copyright holder: Wen et al.
License: This is an open access article distributed under the terms of the Creative Commons Attribution License, which permits unrestricted use, distribution, reproduction and adaptation in any medium and for any purpose provided that it is properly attributed. For attribution, the original author(s), title, publication source (PeerJ) and either DOI or URL of the article must be cited.
License URL: https://creativecommons.org/licenses/by/4.0/

Keywords: Atrial fibrillation, ceRNA network, miRNA, lncRNA, circRNA

Funding: Jiangsu Provincial Medical Innovation Team CXTDB2017015 Jiangsu Commission of Health, China H201665 Six Talent Foundation of Jiangsu Province, China WSN-20 The study was supported by the Jiangsu Provincial Medical Innovation Team (Grant No. CXTDB2017015), the Jiangsu Commission of Health, China (Grant No. H201665) and the Six Talent Foundation of Jiangsu Province, China (Grant No. WSN-20). The funders had no role in study design, data collection and analysis, decision to publish, or preparation of the manuscript.

==============================
Atrial fibrillation (AF) is a prevalent arrhythmia that requires effective biomarkers and therapeutic targets for clinical management. In recent years, non-coding RNAs (ncRNAs) have emerged as key players in the pathogenesis of AF, particularly through the ceRNA (competitive endogenous RNA) mechanism. By acting as ceRNAs, ncRNAs can competitively bind to miRNAs and modulate the expression of target mRNAs, thereby influencing the biological behavior of AF. The ceRNA axis has shown promise as a diagnostic and prognostic biomarker for AF. This review provides a comprehensive overview of the roles of ncRNAs in the development and progression of AF, highlighting the intricate crosstalk between different ncRNAs in AF pathophysiology. Furthermore, we discuss the potential implications of targeting the circRNA/lncRNA-miRNA-mRNA axis for the diagnosis, prognosis, and therapeutic intervention of AF.

Introduction

The incidence and mortality rates of AF are continuously increasing, leading to serious complications such as heart failure and stroke (Fig. 1) (Sagris et al., 2021). The occurrence and development of AF involve different mechanisms and interactions (Kornej et al., 2020). AF often progresses from paroxysmal to persistent (Nattel & Dobrev, 2016). The occurrence and progression of AF are associated with atrial remodeling, which includes electrical remodeling and structural remodeling. Early stages are typically characterized by electrical remodeling, while late stages are characterized by structural remodeling, including fibrosis of atrial muscles and extracellular matrix, amyloidosis, cell apoptosis, and other tissue structural changes (Karam et al., 2017). Increasing evidence suggests that ncRNAs play a crucial role in AF fibrosis and regulate key molecular processes (Franco, Aranega & Dominguez, 2020). Many studies have revealed different expression profiles of ncRNAs in AF and identified related competing endogenous RNA (ceRNA) networks (Fig. 2), which may serve as therapeutic targets for AF and also as biomarkers for AF-related conditions (Liu et al., 2021).

Figure 1 Risk factors for atrial fibrillation.

Figure 2 CeRNA mechanism diagram.

NcRNA is a class of RNA molecules that do not encode proteins but play important biological functions at the RNA level (Winkle et al., 2021). Recent studies have found that miRNA, lncRNA, and circRNA are closely associated with the occurrence and development of AF and play important regulatory roles in transcription, post-transcription, and translation, making them potential biological targets for the prevention, diagnosis, and treatment of AF (van den Berg et al., 2017; Jiang et al., 2019; Cao et al., 2019). Among them, circRNA and lncRNA can act as targets of miRNA, altering the expression levels of mRNA by regulating the levels of miRNA. One miRNA can affect multiple target genes, and multiple miRNAs can affect the same target gene. When different ncRNAs interact with miRNA, miRNA can also act on different target genes. Understanding this novel RNA crosstalk can help researchers better understand gene regulatory networks and their impact on human development and diseases (Tay, Rinn & Pandolfi, 2014). In this review, we provide an overview and discussion of the latest mechanisms mediated by miRNA, lncRNA, and circRNA in the pathogenesis and development of AF (Fig. 3). A deeper understanding of the roles of these three types of RNA can provide unique opportunities for the diagnosis and treatment of AF. The literature we have collected here will be helpful for researchers studying AF as well as ncRNA, ceRNA, and genetics.

Figure 3 Signaling pathway of ncRNA in atrial fibrillation.

Survey methodology

We conducted a literature search using the PubMed database (https://pubmed.ncbi.nlm.nih.gov/). Initially, we used “miRNA and AF”, “circRNA and AF”, and “lncRNA and AF” as search terms. Subsequently, we further refined the search using “lncRNA and miRNA and AF” and “circRNA and miRNA and AF”. Finally, we specified the search terms as “lncRNA, miRNA, and mRNA in AF” and “circRNA, miRNA, and mRNA in AF”. These search terms were used to identify research articles related to miRNA, lncRNA, and circRNA in AF.

miRNA-mediated mechanism in AF

miRNA is a type of endogenous small non-coding RNA that can regulate the expression of corresponding target genes and thus participate in most biological processes. miRNA-mediated gene regulation is accomplished through post-transcriptional inhibition of the small RNA seed region, which binds to the three UTRs of the target mRNA. These effects contribute to the formation of a regulatory network of miRNAs on their targets. Many studies have found that miRNAs play an important role in the pathogenesis of AF (Pu et al., 2019). Some miRNAs have been implicated in the progression of AF by participating in atrial fibrosis (Table 1) (Fig. 4). For example, overexpression of miR-27b-3p can regulate the Wnt/β-Catenin signaling pathway by targeting Wnt3a, thereby influencing the activity of the Wnt/β-Catenin pathway to reduce atrial fibrosis and subsequently decrease the incidence and duration of AF (Lv et al., 2019). Another study has shown that miR-205 improves atrial fibrosis by negatively regulating the expression of P4HA3 and interfering with the fibrosis-related JNK pathway (Xiao et al., 2021a). H2S can upregulate the expression of miR-133a in cardiomyocytes, and overexpression of miR-133a can inhibit Ang II-induced cardiomyocyte proliferation and migration, as well as reduce the levels of fibrosis markers and CTGF (Su et al., 2021). Hsa-miR-4443 targets THBS1 and regulates the TGF-β1/α-SMA/collagen pathway, thereby inhibiting HCFB proliferation and collagen synthesis, leading to a reduction in myocardial fibrosis (Xiao et al., 2021b). Some researchers suggest that miR-21 has the potential to predict AF, particularly in combination with NT-proBNP (Sieweke et al., 2020). Quercetin (Que) can promote the expression of miR-135b, resulting in the downregulation of miR-135b target genes TGFBR1 and TGFBR2, which inhibits the activity of the TGF-β/Smads pathway, thereby reducing myocardial fibrosis and collagen deposition (Wang et al., 2021a). Additionally, quercetin has been shown to inhibit the expression of miR-223-3p, increase FOXO3 expression, and activate related autophagy pathways to prevent myocardial fibrosis (Hu et al., 2021). Overexpression of miR-27b has been reported to inhibit the Smad-2/3 signaling pathway by negatively regulating ALK5, thereby attenuating myocardial fibrosis (Wang et al., 2018). Downregulation of miR-10a can target BCL6 and block the TGF-β1/Smads signaling pathway to reduce atrial structural remodeling (Li et al., 2019). Overexpression of miR-29b can improve atrial fibrosis by targeting and regulating the expression of TGFβRΙ, thereby inhibiting the activity of the Smad-2/3 pathway (Han et al., 2022). Downregulation of miR-138-5p can reverse cardiac fibrosis by targeting and inhibiting CYP11B2 (Xie, Fu & Xie, 2018). MiR-29b-3p negatively regulates the PDGF-B signaling pathway to inhibit atrial remodeling (Lv et al., 2021). Upregulation of miR-205-5p negatively regulates EHMT2, which affects the inhibitory role of IGFBP3 in atrial fibrosis (Xiao et al., 2022). MiRNA-148a alleviates cardiomyocyte apoptosis in AF by inhibiting SMOC2 (Zhang, Man & Chen, 2022).

Table 1 miRNAs inhibiting the occurrence and development of AF.

miRNA	Expression
level	Gene target	Function	Result	
miR-27b-3p	Up	Wnt3a	Wnt/β-Catenin pathway	Inhibit atrial fibrosis	
miR-205		P4HA3	JNK pathway	Inhibit atrial fibrosis	
miR-133a	Up	CTGF	Inhibition of cardiomyocyte proliferation	Inhibit atrial fibrosis	
hsa-miR-4443	Up	THBS1	TGF-β1/α-SMA /collagen pathway	Inhibit atrial fibrosis	
miR-135b	Up	TGFBR1
TGFBR2	TGF-β/Smads pathway	Inhibit atrial fibrosis	
miR-223-3p	Up	FOXO3	Activation of related autophagy pathways	Inhibit atrial fibrosis	
miR-27b	Up	ALK5	Smad-2/3 pathway	Inhibit atrial fibrosis	
miR-10a	Down	BCL6	TGF-β1/Smads pathway	Inhibit atrial fibrosis	
miR-29b	Up	TGFβRΙ	Smad-2/3 pathway	Inhibit atrial fibrosis	
miR-138-5p	Down	CYP11B2		Inhibit atrial fibrosis	
miR-29b-3p	Up		PDGF-B pathway	Inhibit atrial fibrosis	
miR-205-5p	Up	EHMT2
IGFBP3		Inhibit atrial fibrosis	
miR-148a		SMOC2		Inhibit myocardial apoptosis	

Figure 4 miRNA schematic diagram affecting and inhibiting atrial fibrosis development.

In addition to the inhibitory effects of the aforementioned miRNAs, some miRNAs can also promote myocardial fibrosis (Table 2) (Fig. 5). Research has shown that upregulated miR-146b-5p is positively correlated with the expression of fibrosis markers MMP9, TGFB1, and COL1A1, and it can target and downregulate TIMP4, thereby promoting AF fibrosis (Ye et al., 2021). On one hand, CADM1 has been reported as a potential target of miR-21. Upregulated miR-21 can decrease CADM1 expression, which in turn affects STAT3 expression. STAT3 can induce activation and proliferation of cardiac fibroblasts through TGF-β1, thereby promoting cardiac fibrosis. This suggests that miR-21 is an important signaling molecule in cardiac fibrosis remodeling and AF (Cao, Shi & Ge, 2017). On the other hand, research has shown that overexpression of miR-21 can target WWP-1 to promote TGF-β1/Smad2 signaling pathway and induce proliferation of cardiac fibroblasts (Tao et al., 2018). Similarly, in patients with diabetes and AF, the miR-21 subtype miR-21-3p targets FGFR1, regulating FGFR1/FGF21/PPARγ and further promoting adipose browning, thereby exacerbating myocardial fibrosis (Pan et al., 2021). Soeki et al. (2016) confirmed that plasma levels of miR-328 in AF patients were higher than those in the control group, and the plasma levels of miR-328 in the left atrial appendage (LAA) were positively correlated with the LA voltage zone index, suggesting that miR-328 may be involved in the atrial remodeling process in AF patients. Additionally, miR-455-5p can bind to the target gene SOCS3, activating the STAT3 signaling pathway and accelerating the progression of AF (Li et al., 2021a). MiR-1202 can negatively regulate nNOS expression to activate the TGF-β1/Smad2/3 pathway and promote fibrosis (Xiao et al., 2021c). TGF-β can downregulate the expression of Sema3A through the upregulated miR-181b, thereby inducing EndMT and increasing the degree of atrial fibrosis through the LIMK/p-cofilin signaling pathway (Lai et al., 2022). MiR-23 promotes AF progression by positively regulating TGF-β1 (Yu et al., 2019). IL-6 upregulates miR-210, which targets Foxp3 to inhibit Treg function and promote atrial fibrosis (Chen et al., 2020). Research has shown that overexpression of miR-124-3p can promote fibroblast proliferation by negatively regulating AXIN1 in the WNT/β-catenin signaling pathway (Zhu et al., 2022). Furthermore, overexpression of miR-23b-3p and miR-27b-3p target TGF-β1 and TGFBR3, respectively, and induce atrial fibrosis through the activation of the Smad3 signaling pathway (Yang et al., 2019). A recent study has shown that downregulation of miR-425-5p can promote atrial remodeling by negatively regulating CREB1 (Wei et al., 2022).

Table 2 miRNAs promoting the occurrence and development of AF.

miRNA	Expression
level	Gene target	Function	Result	
miR-146b-5p	Up	TIMP4
MMP9	Activation of TGF-β1 pathway	Promote atrial fibrosis	
miR-21	Up	CADM1
STAT3	Activation of TGF-β1 pathway	Promote atrial fibrosis	
miR-21	Up	WWP-1	Activation of TGF-β1/Smad2 pathway	Promote atrial fibrosis	
miR-21-3p	Up	FGFR1	Fat browning	Promote atrial fibrosis	
miR-328					
miR-455-5p	Up	SOCS3	Activate STAT3 pathway	Promote atrial fibrosis	
miR-1202	Up	nNOS	Activation of TGF-β1/
Smad2/3 pathway	Promote atrial fibrosis	
miR-181b	Up	Sema3A	LIMK/p-cofilin pathway	Promote atrial fibrosis	
miR-23		TGF-β1		Promote atrial fibrosis	
miR-210	Up	Foxp3	Inhibiting Treg function	Promote atrial fibrosis	
miR-124-3p	Up	AXIN1	Wnt/β-catenin pathway	Promote atrial fibrosis	
miR-23b-3p
miR-27b-3p	Up	TGF-β1
TGFBR3	Activate Smad3 pathway	Promote atrial fibrosis	
miR-425-5p	Down	CREB1		Promote ventricular remodeling	
miR-155	Up	CACNA1C	L-type ca2+ density	Promote electrical remodeling	
miR-106b-25	Down	RyR2
ATP2A2	Ca2+ homeostasis disorder	Promote electrical remodeling	

Figure 5 Diagram of miRNAs that influence and promote the development of atrial fibrosis.

In addition to their role in myocardial fibrosis, certain miRNAs can also influence the electrical remodeling of AF. For example, elevated expression levels of miR-155 in AF patients can lead to a decrease in the expression of CACNA1C, resulting in a reduction in L-type Ca2+ density and contributing to AF electrical remodeling (Wang et al., 2021b). Furthermore, downregulation of miR-106b-25 in AF patients has been shown to increase the expression of RyR2 and mediate intracellular Ca2+ elevation, thereby participating in the occurrence and development of AF (Zhu et al., 2019).

Based on these studies, it is evident that miRNAs have broad potential in the diagnosis and treatment of AF.

lncRNA-mediated mechanisms in AF

Long non-coding RNAs (lncRNAs) play crucial roles in various key biological processes (Table 3) (Fig. 6). They are essential for controlling cellular biological processes associated with a wide range of human diseases (Ma, Bajic & Zhang, 2013). For instance, LICPAR has been found to regulate atrial fibrosis through the TGF-β/Smad pathway, providing a potential treatment strategy for AF (Wang et al., 2020). Additionally, NRON(lncRNA) has been shown to alleviate atrial fibrosis by inhibiting M1 macrophage activation in atrial myocytes (Sun et al., 2019). Furthermore, circulating GAS5 (lncRNA)has been identified as a potential biomarker for AF diagnosis and prognosis. The downregulation of GAS5 occurs prior to left atrial enlargement and can be used to predict the progression and recurrence of AF (Shi et al., 2021a). Moreover, the downregulation of TCONS_00016478 may promote atrial energy metabolism remodeling and contribute to the development of AF by inhibiting the PGC1-α/PPARγ signaling pathway (Jiang et al., 2022). These findings highlight the potential of targeting these molecules for the diagnosis and treatment of AF.

Table 3 lncRNAs involved in the occurrence and development of AF.

lncRNA	Expression
level	miRNA	Gene target	Function	
LICPAR	Up			TGF-β/Smad	
NRON			IL-12	M1 Macrophage polarization	
GAS5	Up		ALK5	Inhibition of cardiomyocyte proliferation	
TCONS_00016478				PGC1-α/PPARγ pathway	
HOTAIR	Up		PTBP1	Improve the stability of Wnt5a	
NRON	Up	miR-23a		M2 macrophage polarization	
KCNQ1OT1		miR-223-3p		AMPK pathway	
LINC00636	Up	miR-450a-2-3p		MAPK1 anti-fibrosis	
TUG1		miR-29b-3p		TGF-β1 pathway	
H19	Up	miR-29a-3p		VEGFA/TGF-β pathway	
MIAT	Up	miR-133a-3p			
MIAT	Up	miR-485-5p	CXCL10	Promote atrial fibrosis, inflammation and oxidative stress	
PVT1		miR-128-3p	Sp1	TGF-β1/Smad pathway	
PVT1		miR-145-5p	IL-16	Promote macrophage M1 polarization	
XIST	Up	miR-214-3p	Arl2	Cardiomyocyte pyroptosis protects myocardium	
NEAT1	Up	miR-320	NPAS2		
KCNQ1OT1	Up	miR-384	CACNA1C		
LINC00472		miR-24	JP2	Effect of RyR2 on SR Ca2 + Release	
TCONS-00106987	Up	miR-26	KCNJ2	Promote electrical remodeling,internal rectifier K current ( I + K1 )	
HOTAIR	Down	miR-613	Cx43		
LOC101928304	Up	miR-490-3p	LRRC2	PGC-1α-dependent mitochondrial abundance	
TCONS_00075467	Down	miRNA-328	CACNA1C	Caion channel	
FAM201A	Down	miR-33a-3p	RAC3	Promote autophagy	

Figure 6 Schematic diagram of lncRNA affecting the occurrence and development of AF.

CircRNA-mediated mechanism in AF

A group of ncRNAs, known as circRNAs, possess regulatory properties. Their gene expression is more stable, and their molecular structure forms closed loops that are resistant to degradation by RNA exonucleases. Increasing evidence suggests that circRNAs may interact with miRNAs through a sequence-driven sponge effect (Kristensen et al., 2019). This circRNA-miRNA network may play a role in the pathological and physiological processes of cardiovascular diseases. Previous studies have shown that certain circRNAs may be involved in the process of cardiac fibrosis, thereby promoting the occurrence and development of AF (Table 4). For example, Gao et al. (2021) reported that has_circ_0004104 promotes cardiac fibrosis by targeting the MAPK and TGF-β pathways, suggesting its potential as a therapeutic regulator and biomarker for AF. Additionally, mmu_circ_0005019 has been identified to inhibit cardiac fibroblast fibrosis and reverse electrical remodeling of cardiomyocytes, demonstrating its protective role in the development of AF and suggesting its potential as a therapeutic target (Wu et al., 2021). However, the mechanisms of action for these circRNAs in AF have only been partially elucidated. Compared to miRNAs and lncRNAs, circRNAs are relatively understudied, and further extensive clinical and basic experiments are needed to explore the underlying mechanisms between circRNAs and AF.

Table 4 circRNAs involved in the occurrence and development of AF.

circRNA	miRNA	Gene target	Function	
hsa_circ_0004104			MAPK, TGF-β pathway	
mmu_circ_0005019	has-miR-208
has-miR-21			
				
				
circFAT1(e2)	miR-298	MYB	Promotevascular smooth muscle proliferation	
circCAMTA1	miR-214-3p	TGFBR1	Reduce myocardial fibrosis	

Interactions among three RNAs

The mediated mechanism of lncRNA and miRNA in AF

There is increasing evidence that lncRNAs can target miRNAs and participate in AF. To better understand the molecular mechanisms underlying AF progression, extensive research has been conducted on the roles of lncRNAs and their potential downstream miRNA regulatory factors. It has been reported that PVT1 (lncRNA) acts as a sponge for miR-128-3p to promote Sp1 expression, thereby activating the TGF-β1/Smad signaling pathway and promoting proliferation of atrial fibroblasts (Cao et al., 2019). Additionally, it has been found that elevated expression of MIAT (lncRNA) can promote myocardial fibrosis by targeting and downregulating miR-133a-3p (Yao et al., 2020). Furthermore, it has been reported that serum-derived EVs containing MIAT can counteract the inhibitory effect of miR-485-5p on CXCL10, thereby exacerbating atrial remodeling and AF (Chen et al., 2021). Another study demonstrated that miR-223-3p plays a key role in AF by targeting KCNQ1OT1 (lncRNA) (Dai et al., 2021a). Moreover, upregulated KCNQ1OT1 regulates AF by modulating the miR-384/CACNA1C axis in response to angiotensin II-induced AF (Shen et al., 2018). In AF patients, increased DNA methylation levels in LINC00472 regulate AF progression through the LINC00472/miR-24/JP2/RyR2 signaling pathway (Wang et al., 2019). Additionally, LINC00636, an anti-fibrotic molecule, improves cardiac fibrosis in AF patients by inhibiting MAPK1 through miR-450a-2-3p (Liu, Luo & Lei, 2021). HOTAIR (lncRNA) enhances the stability of Wnt5a by binding to PTBP1, thereby promoting AF-related myocardial fibrosis (Tan et al., 2022). Furthermore, HOTAIR has a potential target regulatory relationship with miR-613 in AF. HOTAIR functions as a ceRNA to sponge miR-613 and regulate Cx43 expression (Dai et al., 2020). GAS5 inhibits AF cell proliferation by suppressing ALK5, providing a new perspective on the mechanisms underlying AF (Lu et al., 2019). TUG1 (lncRNA) regulates cardiac fibroblast proliferation through the miR-29b-3p/TGF-β1 axis (Guo et al., 2021). Another study suggests that H19 promotes CF proliferation and collagen synthesis by inhibiting miR-29a-3p/miR-29b-3p-VEGFA/TGF-β axis, providing a potential new direction for AF treatment (Guo et al., 2022). Overexpression of NRON targets and inhibits miR-23a, thereby promoting M2 macrophage polarization and alleviating atrial fibrosis (Li, Zhang & Jiao, 2021).

The mediated mechanism of circRNA and miRNA in AF

According to the aforementioned explanations, it is known that circRNAs can interact with proteins or act as miRNA sponges, regulating the expression of upstream genes and participating in the development of diseases. In recent years, circRNAs have become a research hotspot in AF and have shown great potential as biomarkers and therapeutic targets. Relevant studies have compared paroxysmal AF patients with permanent AF patients and found functional crosstalk between circRNA and miRNA in permanent AF, which may be an important factor in disease progression (Zeng et al., 2022). In addition, Zhang et al. (2023) demonstrated that in non-valvular AF, circRNA-related ceRNA networks showed that circRNA may be involved in regulating has-miR-208b and has-miR-21, thereby affecting AF through changes in calcium and myocardial contraction, and providing potential biomarkers for AF (Costa et al., 2019). These research findings provide important scientific evidence for further studying the role of circRNA in AF and its therapeutic potential.

The mediated mechanism of lncRNA, miRNA and mRNA in AF

Upregulated LOC101928304/LRRC2 competitively binds to miR-490-3p, leading to downregulation of miR-490-3p expression (Ke et al., 2022). XIST acts as a ceRNA to inhibit miR-214-3p, resulting in increased expression of Arl2 and attenuated apoptosis of myocardial cells (Yan et al., 2021). NEAT1 negatively regulates miR-320, leading to decreased expression of miR-320, which further inhibits the expression of NPAS2 and suppresses cardiac fibroblast proliferation (Dai et al., 2021b). PVT1 regulates IL-16 expression by targeting miR-145-5p, promoting M1 macrophage polarization and enhancing proliferation of atrial fibroblasts (Cao et al., 2021).

Furthermore, several lncRNAs are involved in AF electrical remodeling. For example, miRNA-328 negatively regulates TCONS_00075467 and subsequently modulates the downstream protein-coding gene CACNA1C to counteract the effects on electrical remodeling (Li et al., 2017). Downregulated FAM201A regulates RAC3 expression by targeting miR-33a-3p, promoting autophagy and reducing L-type calcium channels, thereby increasing the incidence of AF (Chen et al., 2022). TCONS-00106987 sponge miR-26 and further regulates KCNJ2, promoting atrial electrical remodeling during AF. These studies reveal a pathogenic lncRNA-miRNA regulatory network associated with AF, providing potential therapeutic targets for AF treatment (Du et al., 2020).

The mediated mechanism of circRNA, miRNA and mRNA in AF

circFAT1 (e2) is a circular RNA that regulates the expression of MYB by targeting miR-298. This regulation leads to an increase in the expression level of MYB, which promotes smooth muscle cell proliferation and contributes to the development of AF (Shi et al., 2021b). On the other hand, circCAMTA1 acts as a sponge for miR-214-3p and downregulates the expression of TGFBR1, thereby alleviating atrial fibrosis (Zhang et al., 2023). These findings provide insights into the molecular mechanisms underlying the development of AF and offer potential therapeutic targets for intervention.

The mediated mechanism of circRNA/lncRNA-miRNA-mRNA axis in AF

In our previous studies, we have constructed a circRNA/lncRNA-miRNA-mRNA ceRNA network associated with AF. We have identified the interactions between FTX/hsa_circRNA7571-hsa-miR-149-5p-IL-6/MMP9 and circRNA_2773/XIST-hsa-miR-486-5p-CADM1 (Wen et al., 2023). These interactions form a complex regulatory network that is involved in the occurrence and development of AF. While we have provided initial validation of the roles of circRNA and lncRNA in regulating miRNA and mRNA in AF, the pathogenesis of AF remains highly complex. Further research is needed to investigate the specific roles and mechanisms of circRNA and lncRNA in AF.

Discussion

Research has shown that ncRNA plays a crucial role in various diseases. The application of ncRNA in different diseases has expanded our understanding of the mechanisms underlying disease occurrence and progression. MiRNAs are involved in the regulation of gene expression, while circRNA and lncRNA exert their influence on gene expression by modulating the function of miRNAs. Through the utilization of deep learning models and bioinformatics analyses, researchers have revealed the potential mechanisms and associated genes of ncRNA in diverse diseases. For instance, researchers employed an unsupervised deep learning model called VAEMDA to predict the association between miRNAs and diseases, achieving high area under the curve (AUC) values through cross-validation and exploring the role of miRNAs in diseases (Zhang, Chen & Yin, 2019). In addition, Sun et al. (2020) conducted bioinformatics analysis to unveil circRNAs, miRNAs, and target genes implicated in non-small cell lung cancer (NSCLC), constructing an interaction network. They observed significant correlations between genes such as MYLIP, GAN, CDC, and patient prognosis, shedding light on the potential mechanisms of circRNA in NSCLC (Sun et al., 2020). Furthermore, research delving into the involvement of circRNAs and miRNAs in lung adenocarcinoma (LUAD) uncovered the high expression of hyaluronan-mediated motility receptor (HMMR) in LUAD, which correlates with clinical and pathological parameters. HMMR was proposed as a candidate oncogene in LUAD, presenting promising prognostic indicators and potential therapeutic targets (Li et al., 2021b). Another study exploring LUAD unveiled a circRNA-miRNA-mRNA regulatory network, in which differentially expressed circRNAs may impact the occurrence and development of LUAD through the Wnt and Hippo signaling pathways (Zuo et al., 2021). These investigations provide valuable insight into the mechanistic role of ncRNA in the occurrence and progression of diseases, offering avenues for the identification of novel therapeutic targets. It is worth further exploring the involvement of ncRNA in AF and elucidating its mechanisms to facilitate the development of innovative treatment strategies.

AF is regulated by complex molecular networks involving miRNA, lncRNA, circRNA, and mRNA. These ncRNAs interact with each other and participate in the occurrence and development of AF. In particular, lncRNA and circRNA can act as miRNA sponges, exerting important regulatory roles in various signaling pathways, thereby influencing the occurrence and development of AF. In-depth study of the circRNA/lncRNA-miRNA-mRNA axis can help us better understand the mechanisms of AF and have potential applications in the diagnosis and treatment of AF. Although there have been increasing studies exploring the relationship between AF and ncRNAs, the interactions among lncRNA/circRNA, miRNA, and mRNA are highly complex, and the specific mechanisms behind them are not yet clear. Therefore, further prediction of the targets and related pathways of ncRNAs and the development of novel effective drugs targeting these targets are needed in the future. Additionally, extensive basic and animal experiments are required to explore the impact of the interplay between ncRNAs on the mechanisms of AF.

This comprehensive review summarizes the research progress on the circRNA/lncRNA-miRNA-mRNA axis in AF. By integrating existing literature and research findings, we have gained a deeper understanding of the mechanisms of these ncRNAs in the occurrence and development of AF. In particular, we emphasize the regulatory roles of these RNA molecules in important biological processes such as myocardial fibrosis, changes in myocardial cell electrophysiology, and inflammation. One of the features of this review is the systematic analysis of the complex interactions among circRNA, lncRNA, and miRNA. We propose the mechanism of circRNA/lncRNA as miRNA sponges, elucidating their functions and roles in AF. Furthermore, we discuss the importance of the TGF-β/Smad signaling pathway, as well as other signaling pathways such as Wnt/β-Catenin and JNK, in regulating the abnormal expression of ncRNAs in AF. Although there have been numerous studies on miRNA and lncRNA in the field of AF, research progress on circRNA and AF is relatively limited. Through in-depth analysis in this review, we provide insights into the functional network of circRNA in the mechanisms of AF. Additionally, there is limited research on the circRNA/lncRNA-miRNA-mRNA axis, and further exploration of the related mechanisms is needed. We note that current research mainly focuses on cardiac structural remodeling, such as myocardial fibrosis, while research on electrical remodeling is relatively scarce. Therefore, we emphasize the importance of early electrical remodeling, which is often a key factor in AF development. In conclusion, this review is of great significance for revealing the regulatory mechanisms and potential impacts of the circRNA/lncRNA-miRNA-mRNA axis in AF. This comprehensive study helps deepen our understanding of ncRNAs and provides new directions for the development of novel drugs and treatment strategies. We believe that these research advances have important clinical implications for improving patients’ quality of life and enhancing AF management.

Conclusion

The mechanism of miRNA, lncRNA, and circRNA in AF is discussed in this review. Meanwhile, the interaction between lncRNA/circRNA, miRNA, and mRNA and its potential impact on AF is also discussed. The abnormal expression of ncRNA is primarily responsible for the occurrence and progression of AF by promoting myocardial fibrosis. Many studies are currently being conducted on the TGF-β/Smad, Wnt/β-Catenin, and JNK signaling pathways in regulating the abnormal expression of AF-related ncRNA. Although current studies have discovered a variety of AF-related ncRNAs, only a few ceRNAs have been verified because ceRNA research is still in its infancy. As a result, identifying more meaningful ncRNAs associated with AF and mutual regulatory mechanisms between them will be helpful for further elucidating the pathogenesis of AF and developing more effective drugs and therapeutic strategies.

Additional Information and Declarations

Competing Interests

Author Contributions

Data Availability

The authors declare that they have no competing interests.

Jia-le Wen conceived and designed the experiments, performed the experiments, analyzed the data, prepared figures and/or tables, authored or reviewed drafts of the article, and approved the final draft.

Zhong-bao Ruan conceived and designed the experiments, analyzed the data, authored or reviewed drafts of the article, and approved the final draft.

Fei Wang conceived and designed the experiments, authored or reviewed drafts of the article, and approved the final draft.

Yuhua Hu conceived and designed the experiments, analyzed the data, prepared figures and/or tables, authored or reviewed drafts of the article, and approved the final draft.

The following information was supplied regarding data availability:

This is a literature review.

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
