# Peer review of "Progress of circRNA/lncRNA-miRNA-mRNA axis in atrial fibrillation"

_PeerJ, doi:10.7717/peerj.16604_

## Round 0.1 · original submission · Major Revisions

The manuscript has been assessed by two independent reviewers and I strongly suggest addressing the concerns raised by the reviewers before your paper can be considered for publication.

Language revision is highly recommended.

The authors are suggested to cite previous similar studies and provide enough background information and clear rationale behind the study.

Abstract needs restructuring with emphasis on the significance of the current study.

More concentration is required on "Discussion and Conclusion" part, the authors are suggested to elaborate the contribution of the current study to the available literature.

**Language Note:** The review process has identified that the English language must be improved. PeerJ can provide language editing services - please contact us at copyediting@peerj.com for pricing (be sure to provide your manuscript number and title). Alternatively, you should make your own arrangements to improve the language quality and provide details in your response letter. – PeerJ Staff

Reviewer 1 ·

Basic reporting

1. You should revise your English writing carefully and eliminate small errors in the paper to make the paper easier to understand. I hope the author can seriously revise the full text. For example, it is wrong to use ‘Summaryized and prospected’ as a subtitle. Besides, ‘The mediated mechanism of lncRNA、miRNA and mRNA in AF’ also use the wrong character. There are many other similar mistakes.
2. Please rewrite the abstract to highlight the study's contributions. I cannot tell the author's main contributions from the abstract.
3. This review should add more high-level research work related to Progress of circRNA/lncRNA-miRNA-mRNA axis in atrial fibrillation.
4. The review should be comprehensively improved, otherwise it will be difficult to meet the publication requirements of this journal.

Experimental design

see above

Validity of the findings

see above

Additional comments

see above

Annotated reviews are not available for download in order to protect the identity of reviewers who chose to remain anonymous.

Reviewer 2 ·

Basic reporting

Finding new biomarkers and therapeutic targets are really critical for the clinical diagnosis and treatment of AF. This manuscript briefly summarizes the roles of ncRNA in the occurrence and progression of AF, and discusses the effects of crosstalk between different ncRNAs in AF. The authors should revise this manuscript according the following suggestions.
1) There are some grammatical errors in the article, and the expression of some of the content is not clear enough. The authors need to check the manuscript carefully and made corresponding revision.
2) Some references cited in this manuscript have incomplete information. Moreover, the format of cited conference articles in this manuscript is incorrect. The authors need to check the manuscript carefully and made corresponding revision.
3) In the ‘Discussion and Conclusion’ part of your manuscript, the authors should add one paragraph to explain how this review advances this field of research and/or contributes something new to the literature.
4) The manuscript lacks a comprehensive comparison and contrast of different research articles related to the roles of ncRNA in the occurrence and progression of AF. Such a comparison is crucial to provide readers with a more holistic understanding of the field.
5) The authors should describe the research articles related to the roles of ncRNA in the occurrence and progression of AF in more detail. You can refer to these articles (PMID: 31489920, PMID: 33329727, PMID: 34527588 and PMID: 34306017), and you should cite them if these articles help you improve the manuscript.

Experimental design

See above

Validity of the findings

See above

Additional comments

See above

Annotated reviews are not available for download in order to protect the identity of reviewers who chose to remain anonymous.

---

## Round 0.2 · accepted · Accept

The authors have adequately addressed the reviewer's comment and the manuscript is ready for publication.

Reviewer 1 ·

Basic reporting

Authors have made well revisions addressing my advice.

Experimental design

Authors have made well revisions addressing my advice.

Validity of the findings

Authors have made well revisions addressing my advice.

Reviewer 2 ·

Basic reporting

The authors have revised the manuscript according to my suggestions. I accept the manuscript for publication in this journal.

Experimental design

The authors have revised the manuscript according to my suggestions. I accept the manuscript for publication in this journal.

Validity of the findings

The authors have revised the manuscript according to my suggestions. I accept the manuscript for publication in this journal.

Additional comments

The authors have revised the manuscript according to my suggestions. I accept the manuscript for publication in this journal.